# Modified LOS Path Following Strategy of a Portable Modular AUV Based on Lateral Movement

**Xiaoming Wang \***  **and Gaosheng Wu**

School of Mechanical Engineering, Tianjin University of Science and Technology, Tianjin 300222, China;
w1843085882@mail.tust.edu.cn

**\*** Correspondence: wxm@tust.edu.cn; Tel.: +86-1862-265-5132

**Abstract:** The portable modular AUV (Autonomous Underwater Vehicle), named ZFAUV, has the ability to move laterally. Its turning radius becomes smaller as the forward speed decreases. Based on this special maneuverability, a modified LOS (line of sight) path following strategy, integrating basic LOS and lateral movement, is proposed. The main idea of this strategy is to improve the path following performance through cross-track error and heading error. That is to say, the ZFAUV continues to move toward the current waypoint during a survey task. If ZFAUV deviates from the desired path due to disturbances from the wind, waves, current, or other uncertainties, it gradually returns to the desired path under lateral maneuverability. At the same time, in order to reduce overshoot after reaching the current waypoint, an arc transiting strategy and decelerating strategy (if necessary) are adopted. Through this strategy, the path following performance is greatly improved. Based on mathematical modeling, this strategy is simulated with a square path and a triangular path. The same paths are selected in lake experiments. The experimental results are in agreement with the simulation results, which demonstrate the validity of this strategy.

**Keywords:** path following; waypoint; LOS; lateral movement; cross-track error; heading error

---

## 1. Introduction

At present, AUVs are mainly divided into fully actuated and underactuated systems. Due to the limitations of weight and cost, the typical underactuated systems are adopted by most AUVs. The shape of these AUVs is generally streamlined, and most are torpedo-shaped (e.g., the NERC Autosub6000 AUV and REMUS-100 AUV). Different kinds of control systems, such as the X rudder, cross rudder, rudder behind propeller, rudder at front, and vector propulsion systems, have been adopted for use in underactuated AUVs. The problem of these systems is that the steering efficiency is relatively low at low speed [1,2]. The steering efficiency at low speed can be improved by exploiting multiple fixed thrusters for some large AUVs, such as CR-01, CR-02. In recent years, some small AUVs have also exploited multiple thrusters. Four fixed thrusters are used in the X4AUV [3]. The motion of the Fòlaga [4] is obtained through three jet-pumps. The Sparus II AUV has three thrusters (two horizontal and one vertical) [5]. The MARTA AUV [6] is actuated using six fixed propellers (two main propellers on the vehicle tail, two lateral tunnel thrusters and two vertical ones). Seven fixed propellers are adopted by Vu [7,8]. The actuation properties of some existing AUVs are shown in Table 1.

**Table 1.** Actuation properties of some existing AUVs.

| AUVs or Author | Actuation Property |
| --- | --- |
| REMUS-100 | 1 fixed tail thruster, 4 cross rudders |
| X rudder AUV | 1 fixed tail thruster, 4 X rudders |
| Gavia | 1 fixed tail thruster, 4 rudders behind propeller |
| MUN Explorer AUV | 1 fixed tail thruster, 4 X rudders, 2 rudder at front |
| Alistair Palmer | 1 fixed tail thruster, 2 tunnel thrusters |
| Bluefin-12 | vector propulsion system |
| ABE, SENTRY | rotatable thrusters |
| CR-02 | 4 fixed tail thrusters, 2 tunnel thrusters |
| X4AUV | 4 fixed thrusters |
| Sparus II AUV | 3 fixed thrusters |
| MARTA AUV | 6 fixed thrusters |
| Mai The Vu | 7 fixed thrusters |
| Fòlaga | 3 jet-pumps |

Different from the aforementioned AUVs, we developed a small portable modular AUV, named ZFAUV, with four fixed thrusters arranged at the tail and two tunnel thrusters (one horizontal and the other vertical) set at the front. As shown in Figure 1, it weights approximately 25 kg. This vehicle can turn around in-situ, move laterally, and move vertically up and down. This is impossible for common propeller-rudder AUVs [2].

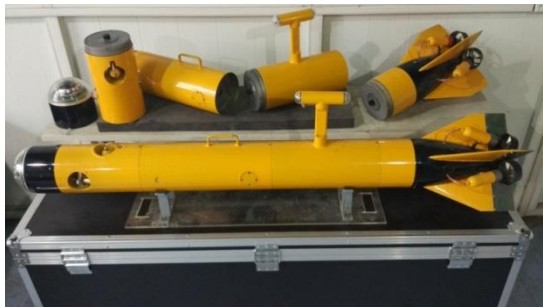

**Figure 1.** Modular ZFAUV.

The maneuverability of ZFAUV was analyzed by Wang [2], and this paper mainly solves the path following problem of ZFAUV.

Path following is one of the typical control scenarios in the control literature, and it pertains to following a predefined path independent of time, i.e., without placing any restrictions on the temporal propagation along the path. This is typical for ships in transit between continents or underwater vehicles used to map the seabed [9,10]. The goal of path-following control is to force an AUV to track a desired path and to make the cross-track error converge to zero quickly and smoothly [11]. For AUVs, tracking a path accurately is an important technical guarantee for their survey tasks (marine mapping, underwater inspection) and their own safety [12,13]. Good path-following performance is a basic performance requirement that ensures AUVs are able to succeed in their underwater survey tasks [14].

There are three kinds of paths in practice: straight line paths among waypoints, dubins and similar paths, and piecewise polynomial and spline paths [15]. For curved paths, the drawback is that the paths must be parametrized and known in advance. In many cases this is not practical, and the simpler path consisting of waypoints and straight lines must be used [9]. The desired path is composed of a collection of waypoints in a waypoint table [16]. AUVs often operate in three-dimensional space underwater. However, it is quite common to assume that altitude/depth is controlled independently such that the path following objective is limited to motion control in the horizontal plane [17]. Thus, the planar straight-line path is considered in this paper.

There are some important guidance laws that are applicable to AUVs [18], including Lyapunov-based guidance, Proportional Navigation Guidance (PNG), and Line-of-Sight (LOS) guidance. Optimal path planning for waypoint guidance of an AUV has been considered [19]. Proportional LOS guidance, proportional-integral LOS guidance [17], and integral LOS [20,21] have been proposed. LOS is the most widely used guidance strategy. In fact, nearly all guidance laws in use today have some form of LOS guidance. In other words, LOS guidance is the key element of most guidance systems [9,22–29].

Two different LOS guidance principles can be used to steer along the LOS vector [23]: enclosure-based steering and lookahead-based steering. The most frequently used method for path following is lookahead-based steering. The main advantages of lookahead LOS guidance are the simplicity and ease of implementation [17]. Some researchers have assumed that a constant lookahead distance [10]. In general, a small lookahead distance will induce more aggressive steering and, thus, the desired path will be reached more quickly, but it might also be the reason for unwanted oscillations around the path. Conversely, a large lookahead distance results in smoother steering, which prevents unwanted oscillations, but the downside is slower convergence to the path [10,30]. With regard to this problem, Lekkas and Fossen proposed a time-varying lookahead distance △ dependent on the cross-track error. This approach results in lower values for △ when the vehicle is far from the desired path and greater △ values when the vehicle is closer to the path, and less abrupt behavior is needed to avoid oscillating around the path [10,27,31]. The disadvantage of this method is that the overshoot at corners is large, so the following performance is poor.

For path following, in addition to the importance of the guidance algorithm, the selection of the next waypoint in the waypoint table is also very important. A common criterion is for the vehicle to be within a circle of acceptance of the current waypoint [9,16,32]. This method is relatively simple, but some waypoints may be missed under certain situations [33].

Based on the maneuverability of ZFAUV and the basic LOS algorithm, a simpler path following algorithm is proposed. The basic idea is to track the desired path with cross-track error and heading error. That is to say, ZFAUV keeps moving toward the current waypoint during survey tasks. If unknown forces (wind, sea currents, wave, et al.) act on common propeller-rudder AUVs, it is impossible (in the general case) to accomplish the motion control task, i.e., to converge to the desired path. Through the lateral movement of ZFAUV, it gradually returns to the desired path and, at the same time, keeps moving toward the waypoint. When following a polygon trajectory, ZFAUV constantly determines whether the current waypoint is reachable or not. If the waypoint is reachable, ZFAUV moves toward this waypoint at survey speed. If the waypoint is unreachable, according to the maneuverability (the faster the speed, the larger the turning radius), ZFAUV decreases the forward speed first and then moves toward the waypoint, without oscillations around the desired path. At the same time, to reduce overshoot after reaching the current waypoint, an arc transiting strategy is adopted. The following performance is greatly improved. The validity of this strategy is verified by simulations and experiments.

The remainder of this paper is organized as follows. Section 2 presents the mathematical model and the motion of ZFAUV in the horizontal plane. The heading keeping strategy is introduced in Section 3. The modified path following strategy based on cross-track error and heading error is given in Section 4. Some experimental results can be found in Section 5. Section 6 concludes the paper.

## 2. Mathematical Modeling

### 2.1. Reference Frames

In this paper, two right-handed reference frames are established: the earth-fixed reference frame $E - X_e Y_e Z_e$ and the body-fixed reference frame $B - xyz$, as shown in Figure 2 [2,34].

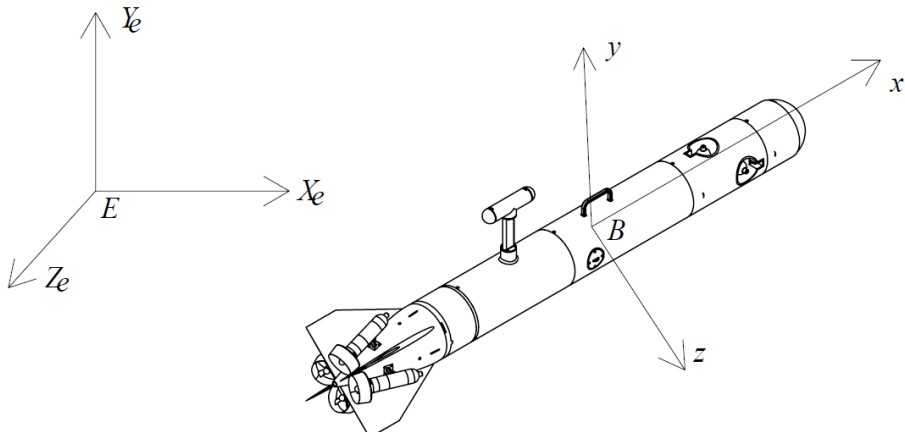

**Figure 2.** Reference frames of ZFAUV.

$E - X_eY_eZ_e$ is fixed with the Earth. The origin can be selected at any position, such as the water surface of the launching point. $EX_eZ_e$ is the horizontal plane, $EX_e$ points toward the direction of launching (e.g., the North), $EY_e$ points upwards normal to the Earth's surface, and $EZ_e$ can be determined by the right hand rule. $B - xyz$ with origin $B$ is a moving reference frame that is fixed to ZFAUV, and $B$ is the center of buoyancy. $Bx$ follows the central line of ZFAUV, which points from aft to fore. When ZFAUV is placed horizontally on the ground, $By$ points upwards normal to the Earth's surface. $Bz$ can be determined by the right hand rule.

## 2.2. Geometric Model and Motion Analysis

The arrangement of the thrusters is shown in Figure 3. $T_5$ is a horizontal tunnel thruster, $T_6$ is a vertical tunnel thruster, $T_1$ and $T_2$ are vertical thrusters, and $T_3$ and $T_4$ are horizontal thrusters. The angle between $T_1$, $T_2$, $T_3$, $T_4$ and the $x$-axis is $\vartheta$, $\vartheta = 22.5°$. The planar straight-line path is considered in this paper, so the situation in the horizontal plane is studied only.

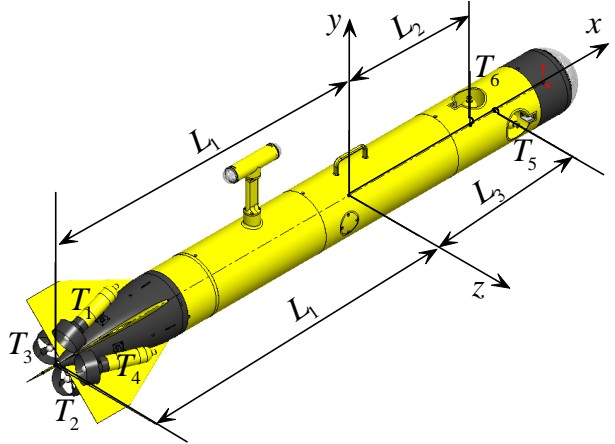

**Figure 3.** Thruster arrangement.

Figure 4 is the top view of ZFAUV. Because the thrust of the thrusters is adjustable, the following simple analysis applies when $|T_3| = |T_4|$ and $T_1 = T_2 > 0$.

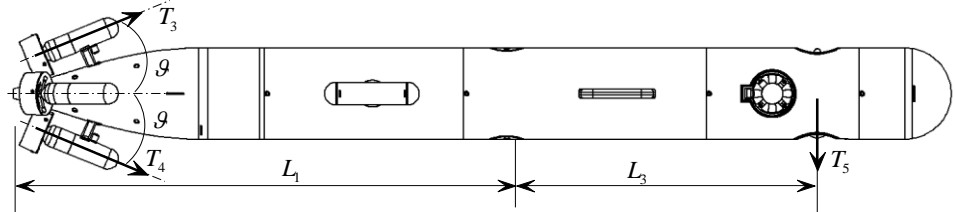

**Figure 4.** Force analysis in the horizontal plane.

(1)  $T_3 = T_4 > 0, T_5 = 0$, ZFAUV moves forward.

(2)  $T_3 = T_4 > 0, T_5 > 0$, ZFAUV turns right.

(3)  $T_3 = T_4 > 0, T_5 < 0$, ZFAUV turns left.

(4)  $T_3 = -T_4 < 0, T_5 > 0, -2T_3 \sin \vartheta L_1 = T_5 L_3$, ZFAUV can move right laterally under certain conditions.

(5)  $T_3 = -T_4 > 0, T_5 < 0, 2T_3 \sin \vartheta L_1 = -T_5 L_3$, ZFAUV can move left laterally under certain conditions.

(6)  $T_3 = -T_4 > 0, T_5 > 0, 2T_3 \sin \vartheta = T_5$, ZFAUV can turn right in-situ under certain conditions.

(7)  $T_3 = -T_4 < 0, T_5 < 0, 2T_3 \sin \vartheta = T_5$, ZFAUV can turn left in-situ under certain conditions.

## 2.3. Modeling in the Horizontal Plane

According to the work of Wang [2,34] and Fossen [9], we designed a simplified model in the horizontal plane as follows.

$$
\begin{cases}
(m + \lambda_{11})\dot{u} = (T_1 + T_2 + T_3 + T_4)\cos\vartheta + \frac{1}{2}\rho V^2{}_T SC_x(0) \\
(m + \lambda_{33})\dot{w} + \lambda_{35}\dot{q} = muq + \Delta G \sin\varphi + \frac{1}{2}\rho V^2{}_T S(C_Z^\beta \beta + C_Z^p p' + C_Z^q q') + T_5 - (T_3 - T_4)\sin\vartheta \\
(J_x + \lambda_{44})\dot{p} = mu y_G q + G y_G \sin\varphi + \frac{1}{2}\rho V^2{}_T SL(C_R^\beta \beta + C_R^p p' + C_R^q q') + (-Q_1 + Q_2 - Q_3 + Q_4)\cos\vartheta \\
(J_y + \lambda_{55})\dot{q} + \lambda_{35}\dot{w} = \frac{1}{2}\rho V^2{}_T SL(C_M^\beta \beta + C_M^p p' + C_M^q q') - (T_3 - T_4)\sin\vartheta L_1 - T_5 L_3 + (-Q_1 + Q_2)\sin\vartheta + Q_6 \\
\dot{\varphi} = p \\
\dot{\psi} = q \cos\varphi \\
\dot{X}_e = u \cos\psi + w \sin\psi \cos\varphi \\
\dot{Z}_e = -u \sin\psi + w \cos\psi \cos\varphi
\end{cases}
\tag{1}
$$

where $\quad$
$$
\begin{cases}
p' = \frac{pL}{V_T} \\
q' = \frac{qL}{V_T} \\
V_T = \sqrt{u^2 + w^2} \\
\beta = \arcsin(\frac{w}{V_T}) \\
u = V_T \cos\beta \\
-(T_1 - T_2)\sin\vartheta L_1 + T_6 L_2 + (Q_3 - Q_4)\sin\vartheta + Q_5 = 0
\end{cases}
, \quad
\begin{cases}
T_1 = \frac{K_{T1}\rho D_1{}^4}{60^2} n_1|n_1|n^2_{\max 1} \\
T_2 = \frac{K_{T1}\rho D_1{}^4}{60^2} n_2|n_2|n^2_{\max 2} \\
T_3 = \frac{K_{T1}\rho D_1{}^4}{60^2} n_3|n_3|n^2_{\max 3} \\
T_4 = \frac{K_{T1}\rho D_1{}^4}{60^2} n_4|n_4|n^2_{\max 4} \\
T_5 = \frac{K_{T2}\rho D_2{}^4}{60^2} n_5|n_5|n^2_{\max 5} \\
T_6 = \frac{K_{T2}\rho D_2{}^4}{60^2} n_6|n_6|n^2_{\max 6}
\end{cases}
,
$$

$$
\begin{cases}
Q_1 = \frac{K_{Q1}\rho D_1{}^5}{60^2} n_1|n_1|n^2_{\max 1} \\
Q_2 = \frac{K_{Q1}\rho D_1{}^5}{60^2} n_2|n_2|n^2_{\max 2} \\
Q_3 = \frac{K_{Q1}\rho D_1{}^5}{60^2} n_3|n_3|n^2_{\max 3} \\
Q_4 = \frac{K_{Q1}\rho D_1{}^5}{60^2} n_4|n_4|n^2_{\max 4} \\
Q_5 = \frac{K_{Q2}\rho D_2{}^5}{60^2} n_5|n_5|n^2_{\max 5} \\
Q_6 = \frac{K_{Q2}\rho D_2{}^5}{60^2} n_6|n_6|n^2_{\max 6}
\end{cases}
$$
, $p$ and $q$ are the angular velocity. $\varphi$ and $\psi$ are the roll angle and heading angle. Details about the meaning of the various parameters can be found in Appendix A.

### 2.4. Maneuverability in the Horizontal Plane

According to the work of Wang [2], we obtained the maneuverability of ZFAUV in the horizontal plane.

The relationship between the turning radius and the speed of the tunnel thruster is shown in Figure 5.

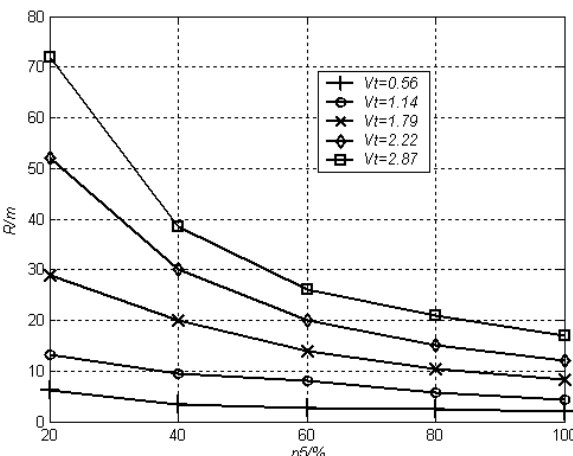

**Figure 5.** Simulation turning radius at different speed.

As seen from Figure 5, at a certain tunnel thruster speed, as the forward speed increases, the turning radius becomes larger. Different from propeller-rudder AUVs, the turning radius is the same at a given rudder angle [2,34].

According to the previous analysis, ZFAUV can move laterally when certain conditions are satisfied. Take right lateral movement as an example, as shown in Figure 6, the equation along *z*-axis can be simplified as follows.

$$
\begin{cases}
\dot{Z}_e = w \\
(m + \lambda_{33})\dot{w} = T_5 - (T_3 - T_4)\sin\vartheta - R_3
\end{cases}
\tag{2}
$$

where $\begin{cases} T_3 = -T_4 \\ R_3 = \frac{1}{2}\rho\omega^2 C_Z S_Z \\ -2T_3\sin\vartheta(L_1 - l_3) = T_5(L_3 + l_3) \end{cases}$ , $R_3$ is the equivalent fluid drag [2]. $l_3$ is the distance of the location of $R_3$ away from the center of buoyancy (*B*). $l_3$ can be obtained by CFD simulation. Figure 7 shows the simulation result. The lateral velocity is approximately 0.4 m/s.

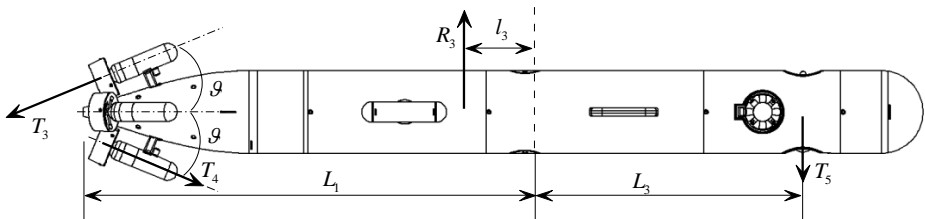

**Figure 6.** Force analysis when moving lateral.

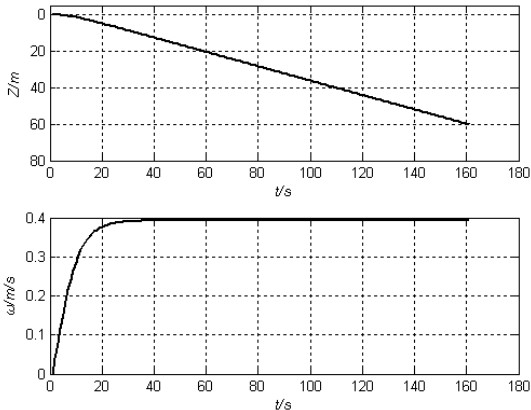

**Figure 7.** Simulation result of lateral movement.

## 3. Heading Keeping Strategy

In many references, the desired heading is assumed to be tracked perfectly at all times [10]. However, tracking the desired heading perfectly is the basis of tracking the desired path. Thus, the heading keeping strategy is introduced here. At the same time, a robust control algorithm is necessary.

To solve the problem of the uncertainty of the system parameters and external disturbance, some control algorithms have been used in AUVs, such as robust adaptive control, sliding mode control and neural network control. An adaptive control law was developed for an AUV to track the desired trajectory [35]. Zhang et al. [36] proposed an adaptive second order sliding mode controller for the AUV path following control.

Fuzzy control is an important branch of intelligent control. Compared with conventional PID control, fuzzy control does not have to establish a mathematical model of the controlled object, and it has the ability to adapt the characteristics of the controlled object, such as time delays, nonlinearities, and time variations. Therefore, the fuzzy PID controller that combines fuzzy control and PID control is adopted for ZFAUV. The structure of the fuzzy PID controller is shown in Figure 8. The fuzzy controller adjusts the PID parameters self-adaptively to satisfy the demand with different *e* and *ec*.

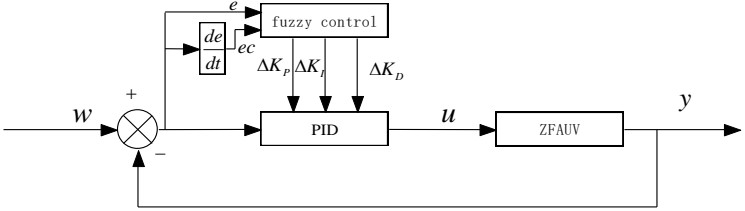

**Figure 8.** Fuzzy PID controller.

The effects of $K_P$, $K_I$, and $K_D$ at different times and the relationships between them must be considered when tuning the PID parameters. The following are the tuning rules with different *e* and *ec*.

(1) When $|e|$ is large, $K_P$ should be large, and $K_D$ should be small for a good following performance. In order to avoid large overshoot, the effect of integration should be limited. Usually, $K_I$ should be zero.

(2) When $|e|$ and $|ec|$ are moderate, $K_P$ should be small for a smaller overshoot. In this case, $K_D$ significantly affects the system. Thus, $K_D$ should be small, and $K_I$ should be moderate.

(3) When $|e|$ is small, both $K_P$ and $K_I$ should be large for better stability. To avoid system oscillation and to consider the anti-interference ability, $K_D$ should be small when $|ec|$ is large, and $K_D$ should be large when $|ec|$ is small.

The ranges of $e$, $ec$, $\Delta K_P$, $\Delta K_I$, and $\Delta K_D$ are defined in the form of a fuzzy set.

$$\{-6, -5, -4, -3, -2, -1, 0, 1, 2, 3, 4, 5, 6\}.$$

The fuzzy subset is given as follows.

$$\{NL, NM, NS, O, PS, PM, PL\}.$$

The membership function is set by a trigonometric function. The membership functions of input and output linguistic variables are shown in Figure 9.

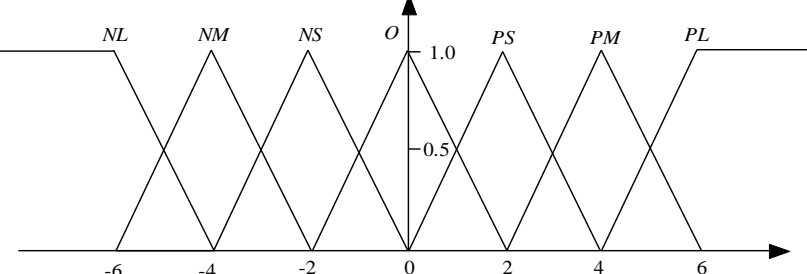

**Figure 9.** Membership functions of the linguistic variables.

According to the tuning rules above, the fuzzy rule of $\Delta K_P$, $\Delta K_I$ and $\Delta K_D$ are shown in Tables 2–4.

**Table 2.** The fuzzy table of $\Delta K_P$.

| e \ ec | NL | NM | NS | O | PS | PM | PL |
|---|---|---|---|---|---|---|---|
| NL | PL | PL | PM | PM | PS | O | O |
| NM | PL | PL | PM | PS | PS | O | NS |
| NS | PM | PM | PM | PS | O | NS | NS |
| O | PM | PM | PS | O | NS | NM | NM |
| PS | PS | PS | O | NS | NS | NM | NL |
| PM | PS | O | NS | NM | NM | NM | NL |
| PL | O | O | NM | NM | NM | NL | NL |

**Table 3.** The fuzzy table of $\Delta K_I$.

| e \ ec | NL | NM | NS | O | PS | PM | PL |
|---|---|---|---|---|---|---|---|
| NL | NL | NL | NM | NM | NS | O | O |
| NM | NL | NL | NM | NS | NS | O | O |
| NS | NL | NM | NS | NS | O | PS | PS |
| O | NM | NM | O | O | PS | PM | PM |
| PS | NM | NS | PS | PS | PS | PM | PL |
| PM | O | O | PS | PS | PM | PL | PL |
| PL | O | O | PM | PM | PM | PL | PL |

**Table 4.** The fuzzy table of $\Delta K_D$.

| e \ ec | NL | NM | NS | O | PS | PM | PL |
|---|---|---|---|---|---|---|---|
| NL | PS | NS | PL | NL | NL | NM | PS |
| NM | PS | NS | PL | NM | NM | NS | O |
| NS | O | NS | NM | NM | NS | NS | O |
| O | O | NS | NS | NS | NS | NS | O |
| PS | O | O | O | O | O | O | O |
| PM | PL | NS | PS | PS | PS | PS | PL |
| PL | PL | PM | PM | PM | PS | PS | PL |

The parameters can be obtained adaptively by the fuzzy controller as follows.

$$
\begin{cases}
K_P = K_P' + \{e_i, ec_i\}_P \cdot k_{\triangle p} \\
K_I = K_I' + \{e_i, ec_i\}_I \cdot k_{\triangle i} \\
K_D = K_D' + \{e_i, ec_i\}_D \cdot k_{\triangle d}
\end{cases}
\tag{3}
$$

where $K_P' = 0.8$, $K_I' = 0.05$ and $K_D' = 1$ are the pre-set values of $K_P$, $K_I$, and $K_D$, which are obtained through the conventional tuning method, and $k_{\triangle p}$, $k_{\triangle i}$, and $k_{\triangle d}$ are the scale factors of $K_P$, $K_I$ and $K_D$, respectively.

In order to obtain a good performance and to make the problem simple, double-loop and closed-loop controls were adopted for heading keeping. The closed-loop control is achieved with a 3D magnetic compass as the feedback and a tunnel thruster as the actuator.

The system includes an inner loop and an outer loop. The heading error $e_\psi$ and the change rate of error $\dot{e}_\psi$ are used as the inputs of the outer loop. The tunnel thruster speed $n_5$ is the output of the outer loop. A motor, a driver and feedback detection elements (an encoder and a hall sensor) constitute the inner tunnel thruster loop by which the high-precision speed control of the tunnel thruster is achieved.

The heading controller is shown in Figure 10.

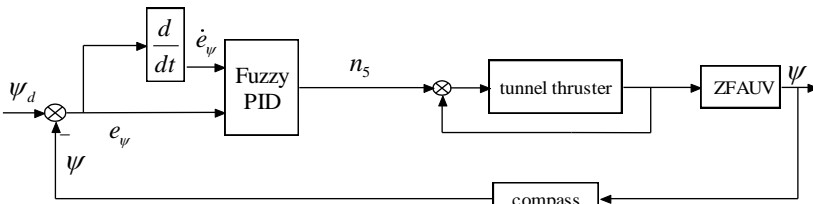

**Figure 10.** Heading controller.

## 4. Path Following Strategy

### 4.1. Problem of Basic LOS Guidance Algorithm

As mentioned in Section 1, basic LOS guidance is a well-documented guidance method given its simplicity and efficiency. With the basic LOS guidance algorithm, the vehicle tries to turn itself directly toward the current waypoint and reach it. The problem is that the waypoints are tracked but the path among them is not, as shown by the dashed LOS vector in Figure 11.

Lookahead-based LOS guidance algorithm is the most frequently used method for path following. With the lookahead-based LOS guidance algorithm, the vehicles try to track the path. Guidance is achieved between two waypoints by inserting a point $p(x_{los}, z_{los})$ that is located on the path between them. The vehicle is then assigned to reach the constantly moving point $p(x_{los}, z_{los})$, as shown by the solid LOS vector in Figure 11.

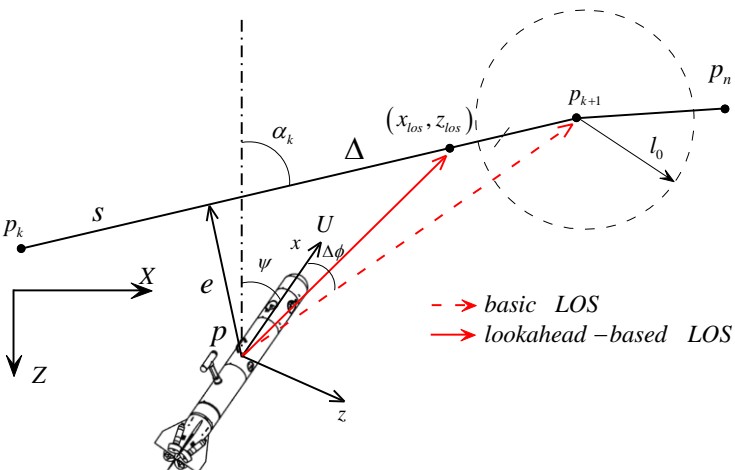

**Figure 11.** Depiction of LOS-based guidance. (*s*, along-track distance; *e*, cross-track error; $\triangle$, lookahead distance; $p_n$, waypoint).

There are some problems for basic LOS and lookahead-based LOS guidance algorithm, as described below, and they should be amended for most practical applications.

1. A common criterion for selecting the next waypoint is for the vehicle to be within a circle of acceptance of the current waypoint [9,16,32]. The radius $\rho_0$ is assumed to be equal to two vehicle lengths, i.e., $\rho_0 = 2L_{vehicle}$, so

$$[x_{k+1} - x(t)]^2 + [z_{k+1} - z(t)]^2 \le \rho_0^2 \tag{4}$$

However, in certain cases, because the turning radius cannot be smaller than the minimum turning radius ($R_0$), it is impossible for underactuated AUVs to accomplish the task of entering the circle of acceptance, as shown in Figure 12. It will always turn around this waypoint, never stop. This waypoint can only be dropped. Another suitable switching criterion [9] solely involves the along-track distance *s*, such that if the total along-track distance between waypoints $p_k$ and $p_{k+1}$ is denoted as $s_{k+1}$, a switch is made when

$$s_{k+1} - s(t) \le \rho_0 \tag{5}$$

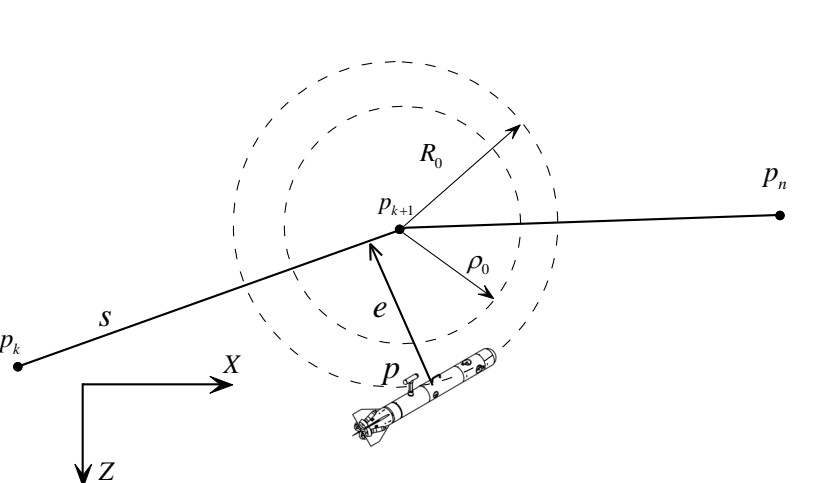

**Figure 12.** Waypoint selecting criterion with along-track distance.

This criterion is similar to Equation (4), but it has the advantage that AUVs do not need to enter the circle of acceptance. The disadvantage is that the current waypoint may be considered has been reached even if the cross-track error is very large, as shown in Figure 12.

2. Overshoot occurs at corners. As mentioned in Section 1, lookahead-based LOS guidance algorithm is the most frequently used method for path following. However, the lookahead distance

has a great influence on the following performance, the simulation results with lookahead-based LOS guidance algorithm are shown in Figure 13. The forward speed is controlled through feedback, $V_T = 2$ m/s. The acceptance radius here is $\rho_0 = 2L_{ZFAUV} = 5$ m. The desired path-1 consists of a total of 4 waypoints: $\{X_1 = 100, Z_1 = 0; X_2 = 100, Z_2 = -100; X_3 = 0, Z_3 = -100; X_4 = 0, Z_4 = 0\}$ (relative to the starting point).

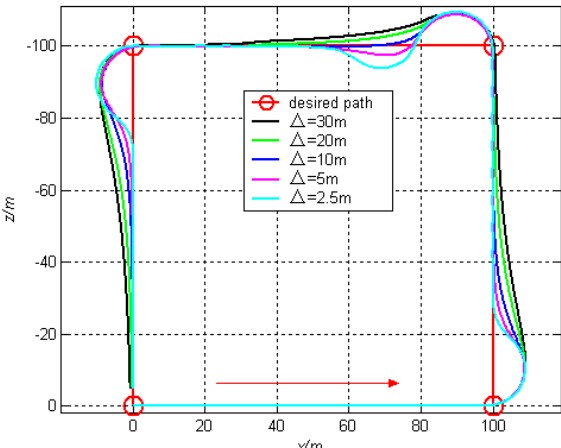

**Figure 13.** Following the square path with lookahead-based LOS.

As seen from Figure 13, the red line is the desired path, a small lookahead distance ($\triangle$) will induce more aggressive steering, a large lookahead distance results in smoother steering, but the downside is slower convergence to the path. $\triangle$ is typically expressed as n vehicle lengths [32]. As seen from Figure 13, the optimal distance is $\triangle = 10$ m, the corresponding simulation result is shown as the blue solid line, there is no unwanted oscillations around the path, and ZFAUV can converge to the desired path. However, no matter $\triangle$ is large, small, or time-varying, there is large overshoot at corners. The maximum overshoot is approximately 8.9 m. When $\triangle = 10$ m, it convergences to the desired path after 40 m. Taking the maximum error of 2 m as the standard, only 75% of the actual trajectory convergences to the desired path, and this number drops to 67% if the first side is not considered (the initial heading angle is the same as that of the first path angle, so there is no error).

The problem becomes more serious when there are sharp corners on the path. So, another desired path-2 consists of a total of 2 waypoints: $\{X_1 = 100, Z_1 = 0; X_2 = 50, Z_2 = -86.6\}$, as shown in Figure 14. The maximum overshoot is approximately 14 m ($V_T = 2$ m/s). Taking the maximum error of 2 m as the standard, only 61% of the actual trajectory convergences to the desired path if only the oblique line is considered.

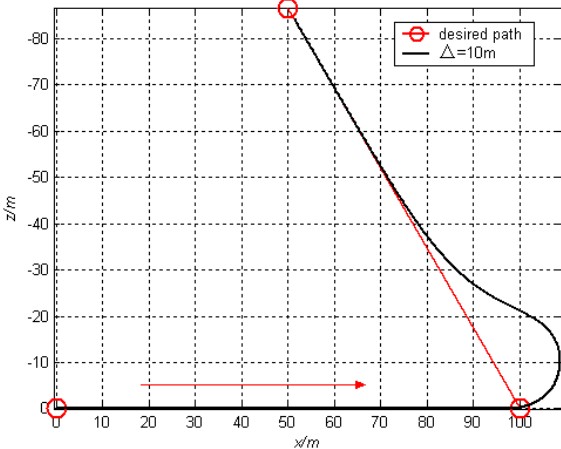

**Figure 14.** Following the triangular path with lookahead-based LOS.

3. Poor following performance occurs if ZFAUV deviates from the desired path due to disturbances from wind, waves, current, or other uncertainties. Take the following situation as an example, shown as in Figure 15. The red line is the desired path, ZFAUV locates at a certain distance (10 m) from the path. The black line is the simulation result of LOS, blue line is the simulation result of lookahead-based LOS ($V_T = 2$ m/s, $\triangle = 10$ m, $\rho_0 = 5$ m). It can be seen from Figure 15a that, with basic LOS, ZFAUV cannot converge to the desired path completely. With lookahead-based LOS, ZFAUV can converge to the desired path basically. However, as can be seen from Figure 15b, the heading angle changes violently with lookahead-based LOS. This will prevent AUVs to accomplish the task like mapping the seabed with acoustic sensors (e.g., multibeam sonar, side-scan sonar, etc.).

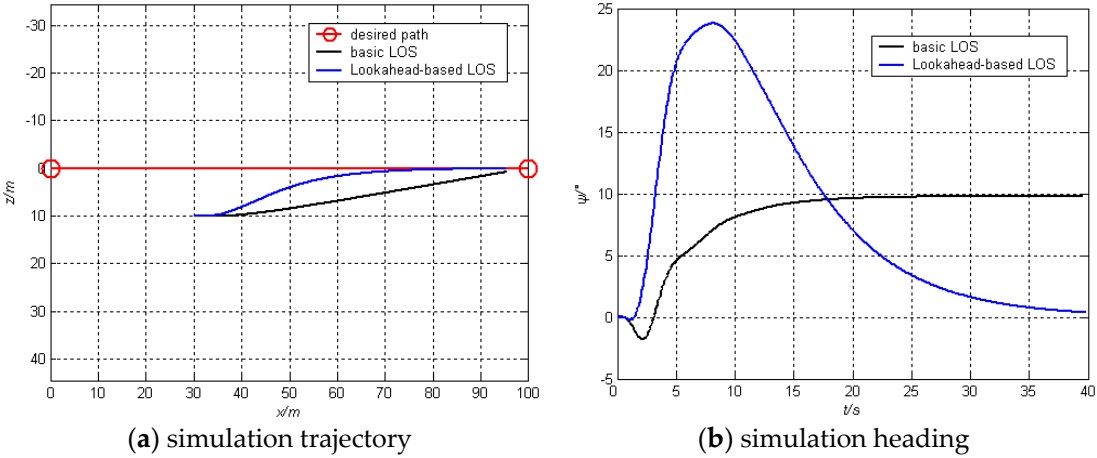

(**a**) simulation trajectory                                    (**b**) simulation heading

**Figure 15.** Path following performance when deviation occurs.

### 4.2. Waypoint Selecting Criteria

As mentioned in Section 4.1, two criteria for selecting the next waypoint have some disadvantages. This paper puts forward a new method. As shown in Figure 16, $R_0$ is the minimum turning radius of ZFAUV at survey speed, and the radius of circles $O$ and $O'$ is $R_0$ also. These circles both pass through the center of buoyancy ($B$), and they are tangent with the path.

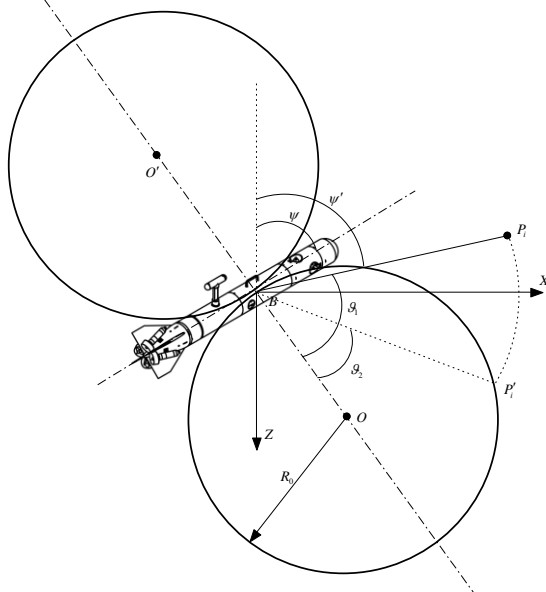

**Figure 16.** Relative position of ZFAUV and the waypoint.

During the survey task, it is impossible to enter the inner area of these circles at survey speed. $\psi$ is the current heading angle. The angle between the line $BP_i$ (which passes ZFAUV and the current waypoint $P_i$) and the North is defined as $\psi'$. The angle between line $BP_i$ and line $OO'$ is defined as $\vartheta_1$. Assuming there is a point $P_i'$ on circle $O$ or circle $O'$ and the length of $BP_i'$ is equal to the length of $BP_i$, the angle between line $BP_i'$ and line $OO'$ is defined as $\vartheta_2$. The relative coordinates between ZFAUV and point $P_i$ are (x, z).

According to the geometric relationship, the following equation is obtained as.

$$\begin{cases} \vartheta_1 = 90° - \psi' + \psi \\ \vartheta_2 = \arccos\left(\frac{BP_i}{2R_0}\right) \end{cases} \tag{6}$$

where $\psi' = \text{atan2}(x, -z)$.

When $BP_i < 2R_0$, $\vartheta_2 \in (-90°, 90°)$, and during the survey task, $\psi \in [-180°, 180°]$. According to the geometric relationship, the range of $\vartheta_1$ can be determined as $\vartheta_1 \in (-360°, 360°)$. $\vartheta$ is defined as

$$\vartheta = \begin{cases} |\vartheta_1| & |\vartheta_1| \in [0°, 90°] \\ 180° - |\vartheta_1| & |\vartheta_1| \in (90°, 180°] \\ |\vartheta_1| - 180° & |\vartheta_1| \in (180°, 270°] \\ 360° - |\vartheta_1| & |\vartheta_1| \in (270°, 360°] \end{cases} \tag{7}$$

If the following condition is satisfied, waypoint $P_i$ is located in circle $O$ or circle $O'$.

$$\begin{cases} BP_i < 2R_0 \\ \vartheta < \vartheta_2 \end{cases} \tag{8}$$

If waypoint $P_i$ locates outside circle $O$ or circle $O'$, in another words, if Equation (8) is not satisfied, it is a reachable point. During the survey task, ZFAUV determines whether the current waypoint is a reachable point or not all the time. If waypoint $P_i$ is a reachable point, ZFAUV will track this point at survey speed continuously. If waypoint $P_i$ is not a reachable point, there are two methods as following.

(1) If the path following requirement is not strict, it is not necessary to track each waypoint accurately, but to cruise at a fixed survey speed, then the current waypoint will be dropped and turn to the next waypoint.

(2) If the path following requirement is strict, and it is necessary to track each waypoint accurately. According to the maneuverability of ZFAUV, as the forward speed decreases, the turning radius becomes smaller and the vehicle can even turn around in-situ, which is the main difference between common propeller-rudder AUVs and ZFAUV. ZFAUV will decelerate the forward speed, so the turning radius will be small until $P_i$ becomes reachable, the heading angle will be adjusted under a smaller turning radius. Then, ZFAUV returns to survey speed. Whether the current waypoint needs to be tracked at survey speed continuously can be decided by the method above.

*4.3. Modified LOS Guidance Algorithm*

As shown in Figures 13 and 14, when the AUV reaches the current waypoint, it should turn to the next waypoint. Because of the existence of the minimum turning radius, it will inevitably deviate from the path. The larger the turning radius, the larger the deviation. When overshoot occurs, it is the same as ZFAUV deviating from the desired path. A lateral movement can be introduced to reduce the overshoot. As shown in Figure 17, considering the ideal situation, the turning radius is assumed to be $R$, forward speed is $V$, then the angular velocity is $\omega = V/R$.

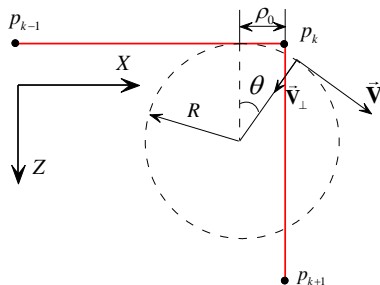

**Figure 17.** Turning at the corner.

Set the time starting to turn to be zero, so the angle at time $t$ is $\theta(t) = \omega t$. The velocity is $\vec{\mathbf{V}} = V_x \vec{\mathbf{x}} + V_z \vec{\mathbf{z}}$, where $\begin{cases} V_x = V \cos(\theta(t)) = V \cos(\omega t) \\ V_z = V \sin(\theta(t)) = V \sin(\omega t) \end{cases}$.

If there is no lateral velocity, $\begin{cases} x = \int V \cos(\omega t) dt \\ z = \int V \sin(\omega t) dt \end{cases}$.

If a lateral velocity is introduced, $\vec{\mathbf{V}}' = \vec{\mathbf{V}} + \vec{\mathbf{V}}_\perp = V_x' \vec{\mathbf{x}} + V_z' \vec{\mathbf{z}}$. where $\begin{cases} V_x' = V \cos(\theta(t)) + V_\perp \cos(\theta(t) + 90) = V \cos(\omega t) - V_\perp \sin(\omega t) \\ V_z' = V \sin(\theta(t)) + V_\perp \sin(\theta(t) + 90) = V \sin(\omega t) + V_\perp \cos(\omega t) \end{cases}$

Then, $\begin{cases} x' = \int (V \cos(\omega t) - V_\perp \sin(\omega t)) dt \\ z' = \int (V \sin(\omega t) + V_\perp \cos(\omega t)) dt \end{cases}$.

Figure 18 shows the simulation result of $V_T = 2$ m/s, $V_\perp = 0.3$ m/s. As seen in Figure 18, the overshoot decreases with the introducing of lateral movement.

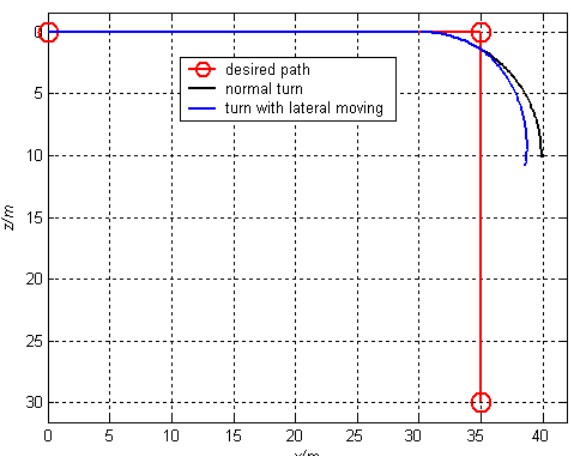

**Figure 18.** Turning at the corner.

The purpose of path following is to make ZFAUV converge to the desired path as far as possible. Therefore, when ZFAUV deviates from the desired path, as shown in Figure 19, the most direct way is to introduce a lateral movement. Under the condition of keeping the heading angle unchanged, the cross-track error can be eliminated.

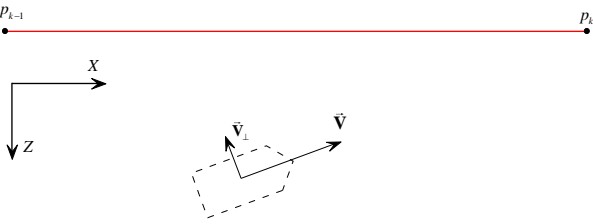

**Figure 19.** Introducing of lateral movement.

Figure 20 shows the simulation result of $V_T = 2\,\text{m/s}$, $V_\perp = 0.3\,\text{m/s}$. It can be seen from Figure 20 that ZFAUV convergences to the desired path rapidly and the change of heading angle is not so intense.

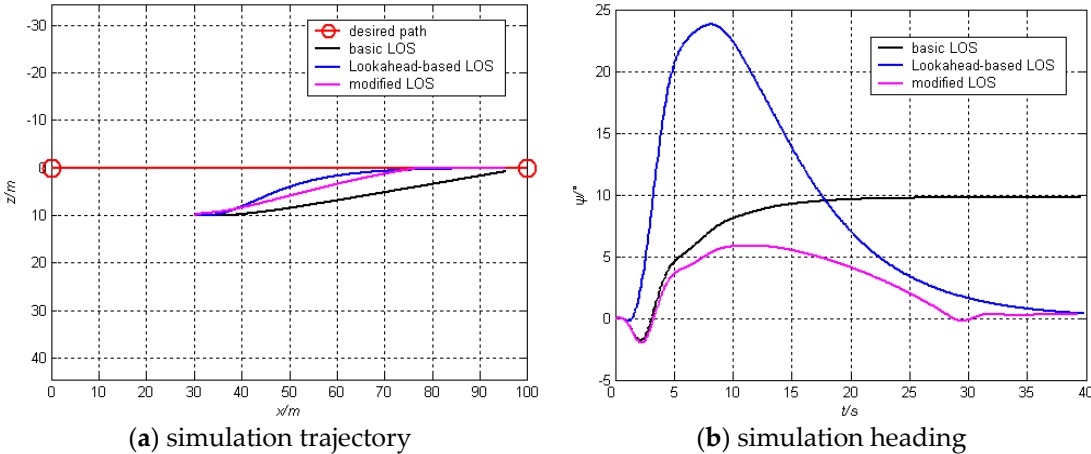

(**a**) simulation trajectory                    (**b**) simulation heading

**Figure 20.** Comparison of different following strategies when deviation occurs.

Therefore, the simpler path following strategy integrating basic LOS and lateral movement is adopted in this paper, as shown in Figure 11, where, the cross-track error and the heading error are the key factors. The nature is to make the cross-track error and the heading error tend to be zero, and ZFAUV moves along with the desired path continuously.

Figure 18 demonstrates the result of fixed forward speed and lateral speed. During the survey task, according to the value of cross-track error, thrusters can be adjusted in real-time, as shown in Figure 11, $e$ is the cross-track error, $\Delta\phi$ is the deviation between heading angle and the view direction. ZFAUV is controlled by $e$ and $\Delta\phi$. The basic idea is to make the trajectory of ZFAUV be close with the desired path as much as possible. The control force is composed of two parts.

$$F = k_e T_e + k_{\Delta\phi} T_{\Delta\phi} \tag{9}$$

where $k_e = \begin{cases} k_1, |e| \geq |e_0| \\ \frac{|e|}{|e_0|} k_1, |e| < |e_0| \end{cases}$ , $k_{\Delta\phi} = \begin{cases} \frac{|e_0|}{|e|} \frac{|\Delta\phi|}{|\Delta\phi_0|} k_2, |e| \geq |e_0| \\ \frac{|\Delta\phi|}{|\Delta\phi_0|} k_2, |e| < |e_0| \end{cases}$ , and $k_1$ and $k_2$ are constants, $T_e$ represent the thruster that moves it laterally ($T_3\ T_4\ T_5$), $T_{\Delta\phi}$ represent the thruster that turns it around ($T_5$).

When ZFAUV is far away from the desired path, the first part of Equation (9) plays a major role in bringing ZFAUV close to the desired path. When the cross-track error is smaller than the given error ($e_0$), the second part of Equation (9) plays a major role in making the heading angle of ZFAUV parallel to the desired path.

During the survey task, the following variables should be computed in real time.

$$\begin{cases} \alpha_k = \text{atan2}[x_{k+1} - x_k, -(z_{k+1} - z_k)] \\ e(t) = [z(t) - z_k]\sin(\alpha_k) + [x(t) - x_k]\cos(\alpha_k) \\ \Delta\phi = \text{atan2}[x_{k+1} - x(t), -(z_{k+1} - z(t))] - \psi \end{cases} \tag{10}$$

The thruster allocation and control strategy under the survey task is shown in Table 5. The lateral velocity is controlled indirectly by controlling the rotation speed and direction of $T_3$, $T_4$ and $T_5$. According to Equation (9), lateral velocity $V_\perp$ is zero when the cross-track error is zero.

**Table 5.** Thruster allocation.

| Different Situations | Illustration | Thruster Allocation |
|---|---|---|
| $e > 0$ $\Delta\phi \geq 0$ | | $T_3 > 0, T_4 = -T_3, T_5 = T_{51} + T_{52},$ $T_{51} = -k_e \frac{T_{5max}}{4}, T_3 = k_e \frac{T_{5max}L_3}{8\sin\vartheta L_1}$ $T_{52} = k_{\Delta\phi} T_{5max}$ |
| $e > 0$ $\Delta\phi < 0$ | | $T_3 > 0, T_4 = -T_3,$ $T_{5temp} = T_{51} + T_{52},$ $T_{51} = -k_e \frac{T_{5max}}{4}, T_{52} = -k_{\Delta\phi} T_{5max},$ $k = \frac{|T_{5temp}|}{|T_{5max}|}$ if $k \leq 1, T_5 = T_{5temp},$ $T_3 = k_e \frac{T_{5max}L_3}{8\sin\vartheta L_1}$ if $k > 1, T_5 = -T_{5max},$ $T_3 = k_e \frac{T_{5max}L_3}{8\sin\vartheta L_1} \frac{1}{k}$ |
| $e < 0$ $\Delta\phi > 0$ | | $T_3 < 0, T_4 = -T_3,$ $T_{5temp} = T_{51} + T_{52},$ $T_{51} = k_e \frac{T_{5max}}{4}, T_{52} = k_{\Delta\phi} T_{5max},$ $k = \frac{|T_{5temp}|}{|T_{5max}|}$ if $k \leq 1, T_5 = T_{5temp},$ $T_3 = -k_e \frac{T_{5max}L_3}{8\sin\vartheta L_1}$ if $k > 1, T_5 = T_{5max},$ $T_3 = -k_e \frac{T_{5max}L_3}{8\sin\vartheta L_1} \frac{1}{k}$ |
| $e < 0$ $\Delta\phi \leq 0$ | | $T_3 < 0, T_4 = -T_3, T_5 = T_{51} + T_{52},$ $T_{51} = k_e \frac{T_{5max}}{4}, T_3 = -k_e \frac{T_{5max}L_3}{8\sin\vartheta L_1}$ $T_{52} = -k_{\Delta\phi} T_{5max}$ |
| $e = 0$ $\Delta\phi \geq 0$ | | $T_3 > 0, T_4 = T_3, T_5 = k_{\Delta\phi} T_{5max}$ |
| $e = 0$ $\Delta\phi < 0$ | | $T_3 > 0, T_4 = T_3, T_5 = -k_{\Delta\phi} T_{5max}$ |

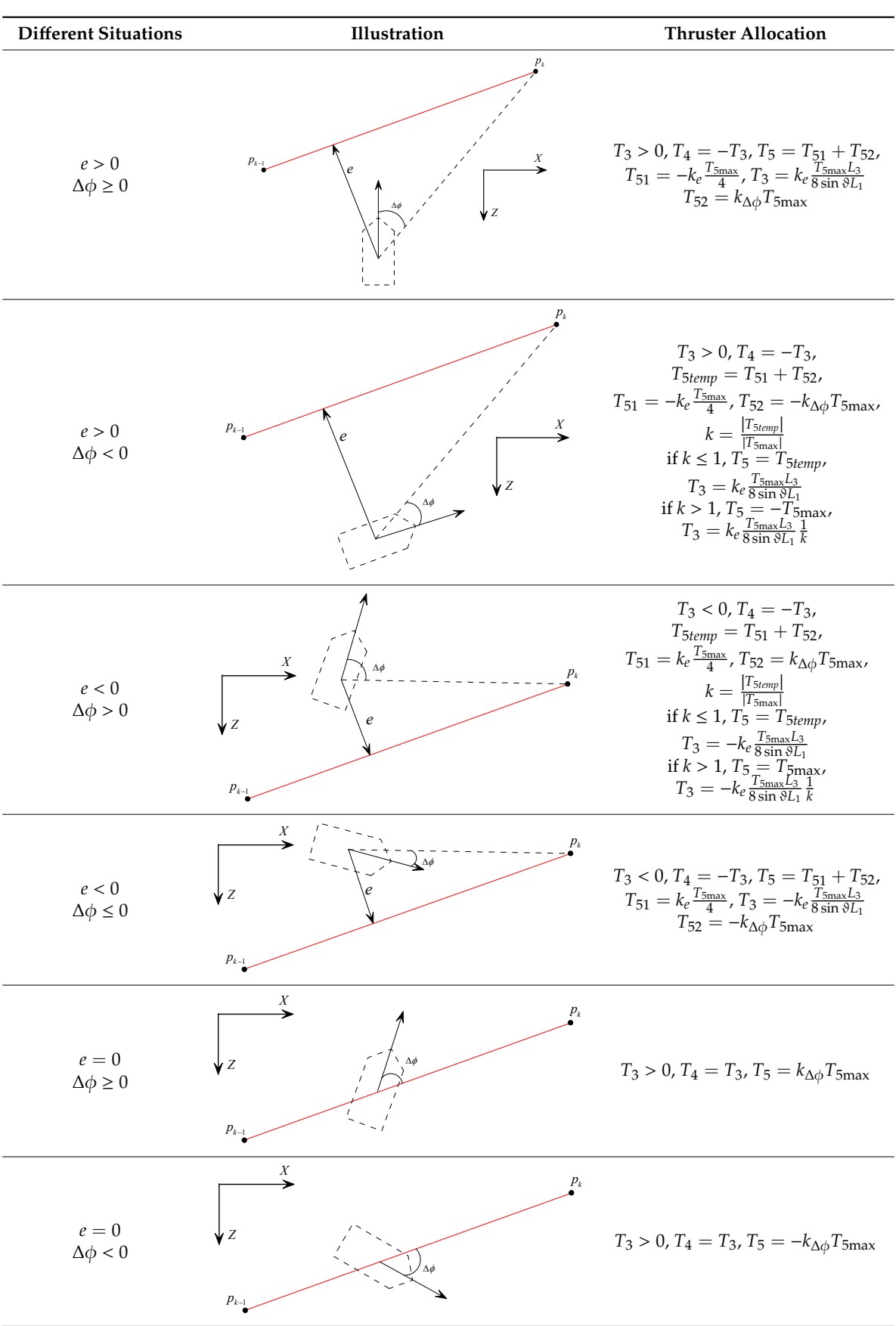

The path following controller is shown in Figure 21.

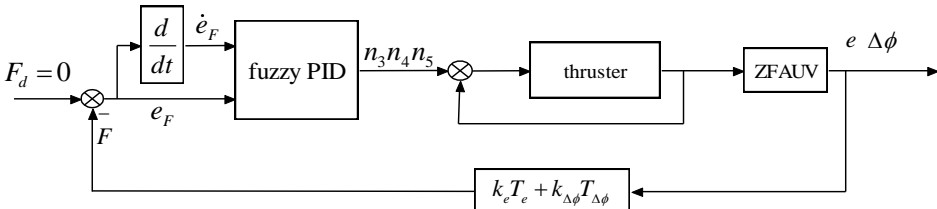

**Figure 21.** Path following controller.

To test the validity of the modified LOS guidance algorithm presented in this section, the same paths were selected. Figure 22 shows the results of the modified LOS ($V_T = 2$ m/s). For path-1, shown in Figure 22a, the following performance is satisfactory at the beginning, and overshoot occurs at the corners. The maximum overshoot is approximately 5.7 m. Taking the maximum error of 2 m as the standard, approximately 84% of the actual trajectory convergences to the desired path, and this drops to approximately 78% if the first side is not considered. For path-2, shown in Figure 22b, the maximum overshoot is approximately 7 m. Taking the maximum error of 2 m as the standard, only 69% of the actual trajectory convergences to the desired path if only the oblique line is considered.

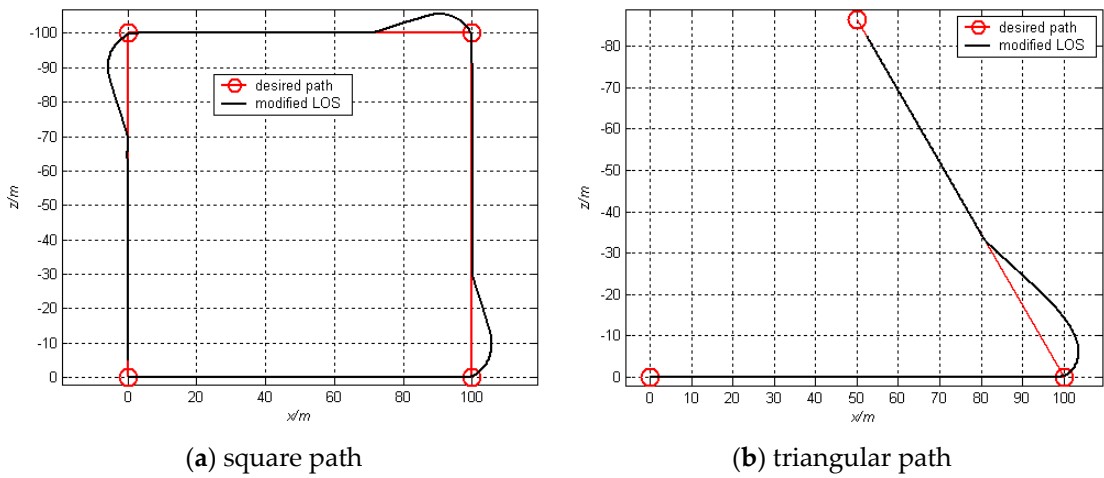

(**a**) square path           (**b**) triangular path

**Figure 22.** Simulation results with modified LOS.

Compared with the lookahead-based LOS algorithm, although the overshoot is reduced, it still exists, especially at the corners. The reason is that the turning radius is too large when ZFAUV moves at survey speed. The problem can be solved by the following method.

### 4.4. Arc Transiting at the Corners

Based on the simulation results above, a larger overshoot exists at the corners still. For the turning radius cannot be smaller than the minimum turning radius. When the acceptance radius is fixed, and if the acceptance radius is too small, overshoot is inevitable. In order to solve this problem, one easy method is to increase the acceptance radius. So, the vehicle will start the curve earlier to avoid the overshoot, as shown in Figure 23 ($V_T = 2$ m/s). However, there are still some problems to be solved.

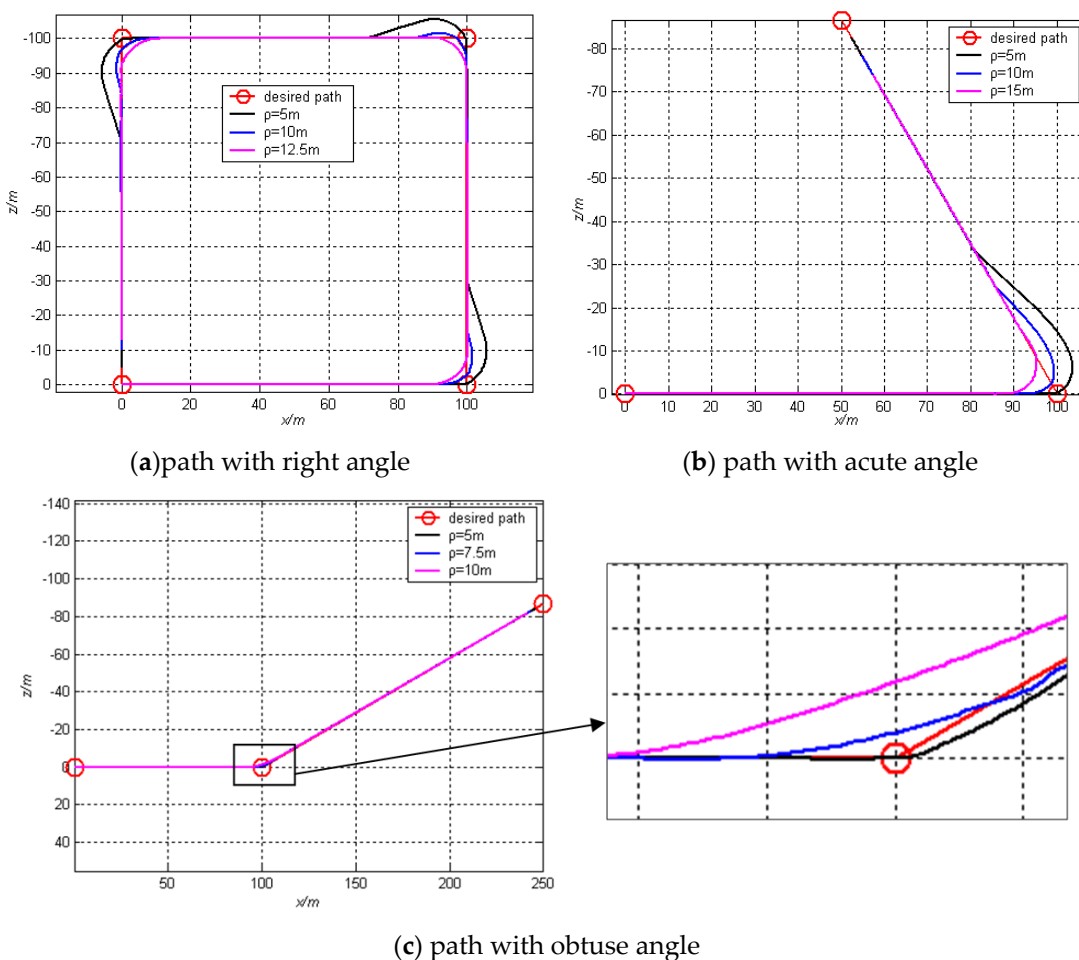

(**a**)path with right angle　　　　　　　(**b**) path with acute angle

(**c**) path with obtuse angle

**Figure 23.** Simulation results with different acceptance radius.

(1) Turning at a large distance from the waypoint, there will be a large arc at corner, and it will stop at a large distance from the last waypoint, so the following performance is getting worse, as shown in Figure 23.

(2) The optimal acceptance radius is different at different corners, as shown in Figure 23.

For right angle (90°), as shown in Figure 23a, the optimal acceptance radius is 12.5 m, the maximum error is approximately 4 m, and 94% of the actual trajectory convergences to the desired path.

For acute angle (60°), as shown in Figure 23b, the optimal acceptance radius is 15 m, the maximum error is approximately 6 m, and 84% of the actual trajectory convergences to the desired path.

For obtuse angle (150°), as shown in Figure 23c, the optimal acceptance radius is 5 m, the maximum error is approximately 0.2 m, and almost all the actual trajectory convergences to the desired path.

(3) Even if all corners are right angles, with the different attitude and position of ZFAUV, the optimal acceptance radius is different also.

So, a fixed acceptance radius is not suitable for all situations. According to the maneuverability of ZFAUV, this paper proposes a strategy to transit to the next path with a fixed acceptance radius (5 m) as following.

As shown in Figure 24, when the current waypoint $P_i$ is tracked by ZFAUV, ZFAUV will move along an arc which is tangential to line $P_iP_{i+1}$ to the position $P_i'$ at a fixed tunnel speed. Then, it will move toward the next waypoint $P_{i+1}$.

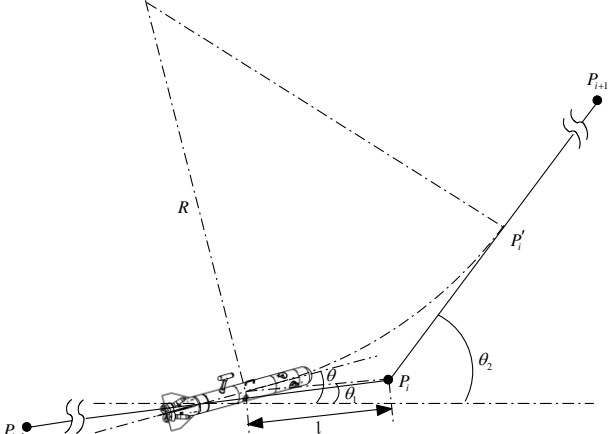

**Figure 24.** Arc transiting strategy at the corner.

As shown in Figure 24, $\theta_1$ is the view direction angle between target point $P_i$ and ZFAUV. $\theta_2$ is the direction angle between waypoint $P_i$ and waypoint $P_{i+1}$. The distance between the current position of ZFAUV and waypoint $P_i$ is defined as $l$, $l$ should be no larger than $\rho_0$. The heading angle is defined as $\theta$. According to the geometric relationship, the transiting arc radius is obtained as

$$R = l\frac{\sin(\theta_2 - \theta_1)}{\sin(\theta_2 - \theta)}\mathrm{ctan}\left(\frac{\theta_2 - \theta}{2}\right) \tag{11}$$

(1) If R ≥ R0, where R0 is the minimum turning radius of ZFAUV at survey speed, according to the relationship between the tunnel speed and the turning radius, the tunnel speed $n_5$ is determined.

(2) If R < R0, according to the maneuverability of ZFAUV (as the forward speed decreases, the turning radius becomes smaller and the vehicle can even turn around in-situ), ZFAUV decelerates the forward speed to ensure R ≥ R0. Then, smooth transition can be achieved. This is impossible for most existing propeller-rudder AUVs.

Another simulation was constructed with the modified LOS and arc transiting at the corners ($V_T = 2$ m/s), and the same paths were selected. For path-1, as seen in Figure 25a, the maximum error is approximately 3 m, and it also occurs at the corners. Taking the maximum error of 2 m as the standard, almost 97% of the actual trajectory coincides with the desired trajectory, and this drops to approximately 96% if the first side is not considered. For path-2, as seen in Figure 25b, the maximum error is approximately 3 m. Taking the maximum error of 2 m as the standard, approximately 95% of the actual trajectory convergences to the desired path if only the oblique line is considered.

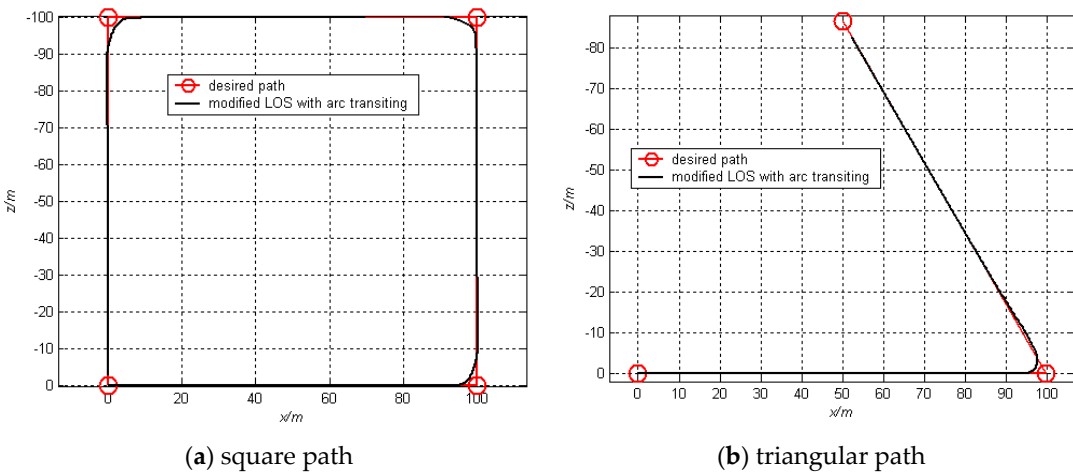

(**a**) square path          (**b**) triangular path

**Figure 25.** Simulation results with the modified LOS and arc transiting at the corners.

Simulation results of the proposed method indicate that smooth convergence and small overshoot without oscillations around the desired path are achieved. Moreover, the problem of some waypoints cannot be reached can be solved completely.

## 5. Experimental Results

A series of experiments were carried out in Daheiting Lake and Qiandao Lake (shown in Figure 26) to verify the performance of the proposed path following strategy. In the lake experiments, the speed is controlled indirectly by controlling the rotation speed and direction of the thrusters.

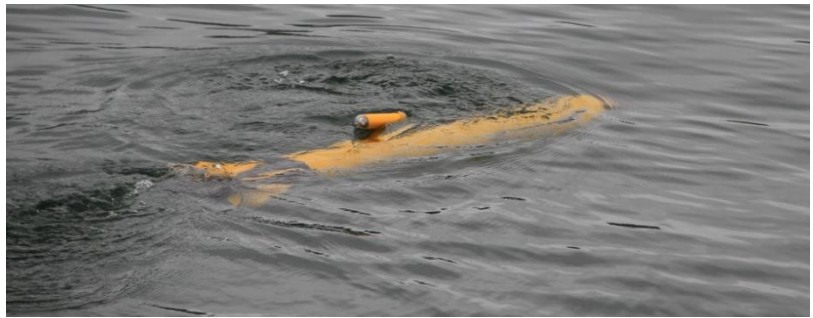

**Figure 26.** ZFAUV in lake.

### 5.1. Basic Experiment

To verify the maneuverability of ZFAUV, turning experiment, heading keeping experiment and lateral moving experiment were carried out first, and these experiments served as the basis for a path following experiment.

Figure 27 shows the result of the turning experiment. In Figure 27a, the forward speed is 80%, the tunnel speed is 20%, and the turning radius is approximately 54 m. In Figure 27b, the forward speed is 80%, the tunnel speed is 100%, and the turning radius is approximately 12.5 m.

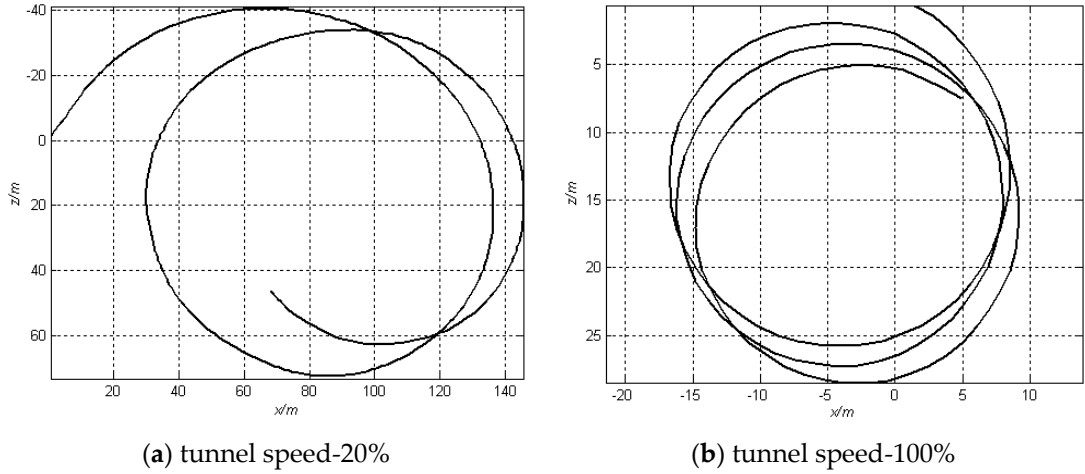

(**a**) tunnel speed-20%     (**b**) tunnel speed-100%

**Figure 27.** Turning experiment.

Table 6 and Figure 28 show the turning radius of ZFAUV at different speed. It can be seen that the experimental results coincide with the simulation results.

**Table 6.** Turning radius of ZFAUV at different speed.

| Tunnel Forward | 20% | 40% | 60% | 80% | 100% |
|---|---|---|---|---|---|
| 0.56 m/s | 6 | 3.5 | 2.7 | 2.5 | 2.2 |
| 1.14 m/s | 13.5 | 10 | 8.5 | 6 | 5 |
| 1.79 m/s | 29.5 | 22.5 | 17.5 | 11 | 8.5 |
| 2.22 m/s | 54 | 30 | 22.5 | 15 | 12.5 |
| 2.87 m/s | 75 | 37.5 | 25 | 20 | 17.5 |

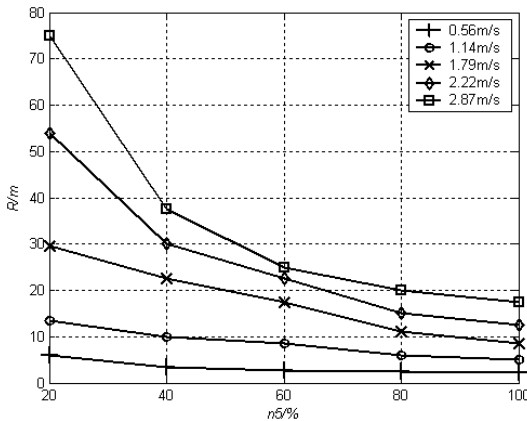

**Figure 28.** Experimental turning radius at different speed.

Figure 29 shows the results of the lateral moving experiment. As seen from Figure 29, the heading oscillates around 147°, and the coordinate of the final point is (30.5, −19.5), the angle of the actual trajectory is approximately 57.4° ($\text{atan2}(x, -z) = \text{atan2}(30.5, 19.5) = 57.4°$). The actual trajectory is basically perpendicular to the heading, and the velocity is approximately 0.38 m/s.

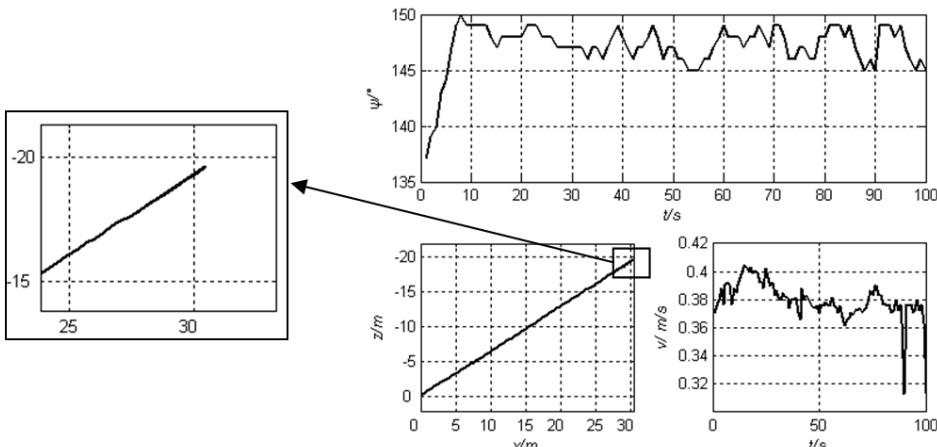

**Figure 29.** Lateral moving.

Figure 30 shows the results of ZFAUV tracking a heading of −5°. As seen from Figure 30, the coordinate of the final point is (−173, −1791), the angle of the actual trajectory is approximately −5.5° ($\text{atan2}(x, -z) = \text{atan2}(-173, 1791) = -5.5°$), the amplitude of the heading oscillations are between −3° and −7°, and the maximum error with the commanded heading is approximately 2°. Therefore, we can conclude that the performance of heading keeping is satisfactory. These results show that the control precision and dynamic performance of the fuzzy PID controller are high enough.

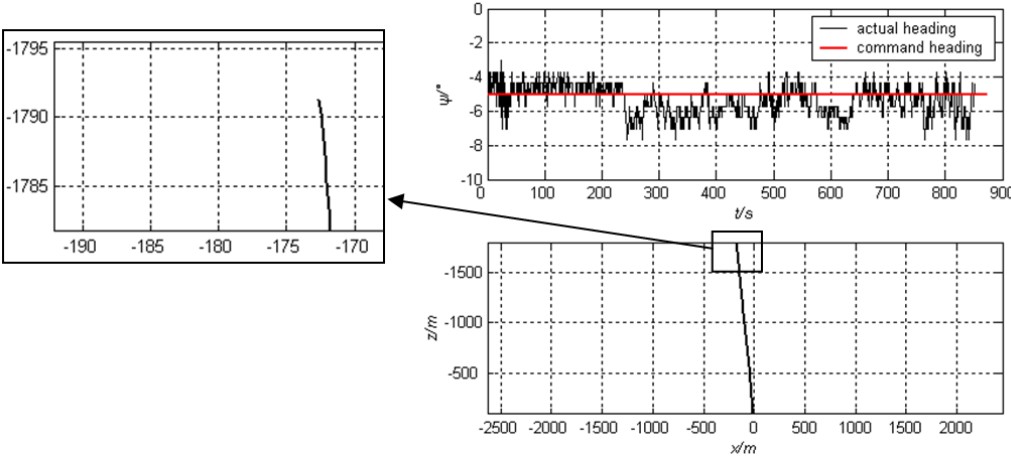

**Figure 30.** Heading keeping with −5°.

## 5.2. Path Following Experiment

In order to verify the performance of different algorithm, lookahead-based LOS guidance algorithm was adopted first ($V_T = 2$ m/s), and the same path-1 was selected. Figure 31 shows the experimental result. It can be seen from the comparison with Figure 14 that the simulation result is in good agreement with the experimental result. Importantly, the overshoot at corner is large.

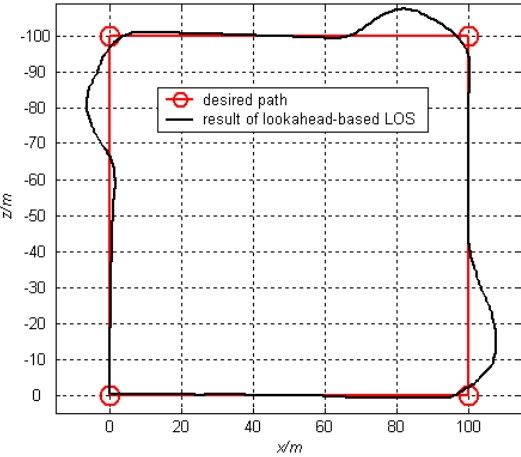

**Figure 31.** Path following experiment with lookahead-based LOS.

Then, the modified LOS guidance algorithm was adopted ($V_T = 2$ m/s), and the same paths were selected. Figure 32 shows the experimental results. For path-1, Figure 32a is the actual trajectory of ZFAUV, Figure 32b is the heading, and Figure 32c is the speed of $T_5$. For path-2, Figure 32d is the actual trajectory of ZFAUV, Figure 32e is the heading, and Figure 32f is the speed of $T_5$. As can be seen from Figure 32 that, the error is quite small but black line does not seem to converge to the red one. The reason is that the experiments were conducted in lake, ZFAUV was disturbed by wind, wave, current and other factors. So, the actual trajectory cannot converge to the desired path completely. Nevertheless, the experimental results are in good agreement with the simulation results, and the overshoot at corner is small.

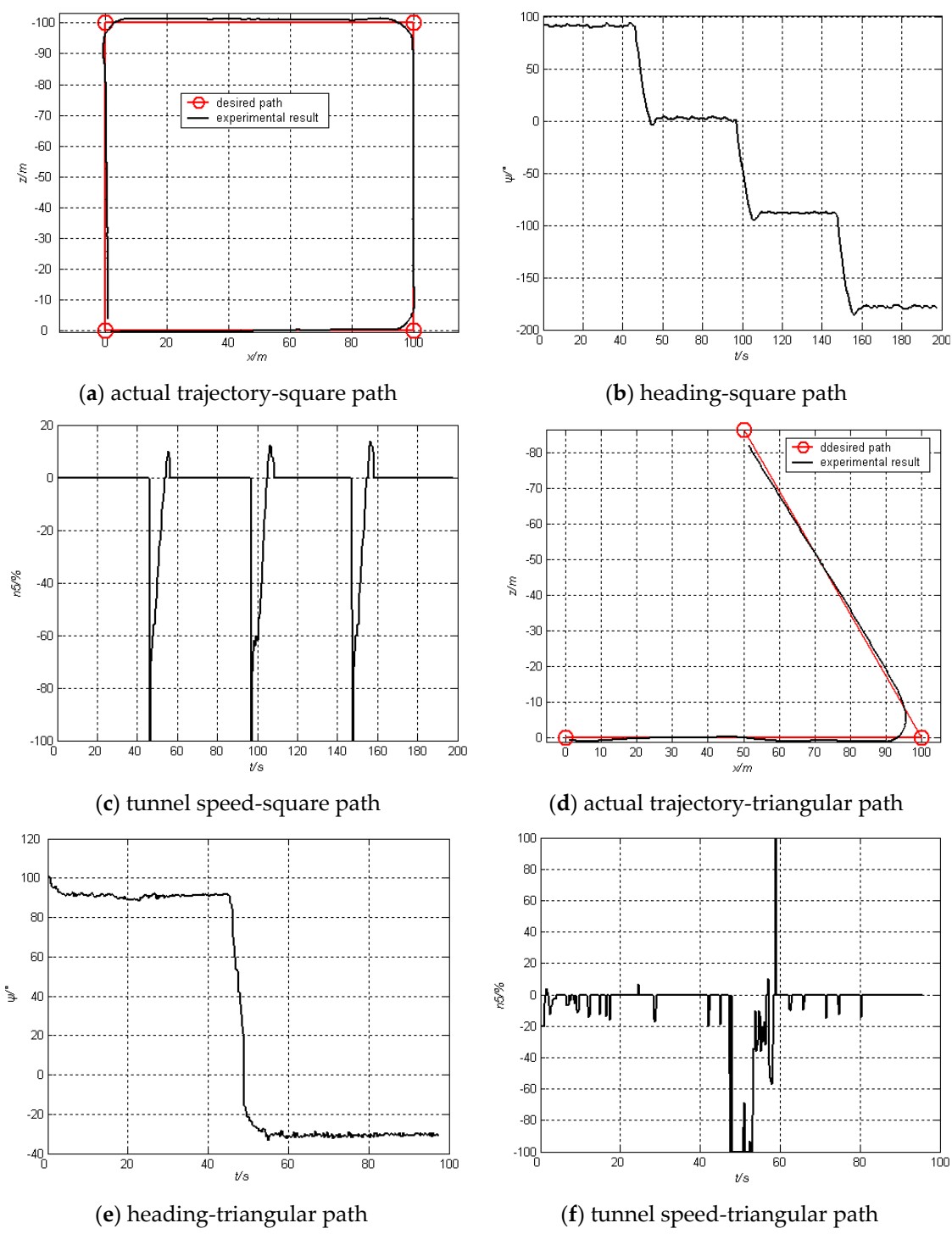

**Figure 32.** Path following experiment with modified LOS.

## 6. Conclusions

ZFAUV has four fixed thrusters at the tail and two tunnel thrusters at the front. It can turn around in-situ and move laterally as well as vertically. In contrast to common propeller-rudder AUVs, the turning radius is related to the forward speed. The smaller the forward speed, the smaller the turning radius.

In order to improve the path following performance, based on the maneuverability of ZFAUV, a modified LOS path following strategy is proposed in this paper. The main motivation is to eliminate cross-track error by lateral movement. ZFAUV continues to move toward the current waypoint,

while the cross-track error can be eliminated with the lateral movement. A method to determine whether the current waypoint is reachable or not is proposed. Smooth transition to the next waypoint is achieved by arc transiting strategy at corners to reduce overshoot. If the calculated transiting radius is smaller than the turning radius under survey speed, decelerating strategy will be adopted to reduce the turning radius, so overshoot at corners can be reduced. In comparison with basic LOS and lookahead-based LOS strategy, the proposed strategy provides better convergence and smaller overshoot. With this strategy, ZFAUV is able to follow less regular paths, e.g., paths with sharp corners. Finally, the simulation results show a satisfactory path following performance. Moreover, the experimental results are consistent with the simulation results. The research findings will be used for the reference or inspiration for improving development of AUVs.

This strategy is based on the maneuverability of ZFAUV, especially as the lateral moving ability and the turning radius can be reduced with a decelerating strategy. At present, this is not suitable for all AUVs, so we will continue to improve this strategy so that it can be adopted by all kinds of AUVs in the future.

**Author Contributions:** Conceptualization, methodology, validation, writing—original draft preparation, X.W.; investigation, writing—review and editing, G.W. All authors have read and agreed to the published version of the manuscript.

**Funding:** This research is supported by the Scientific Research Project of Tianjin Municipal Education Commission (No.2017KJ022).

**Acknowledgments:** We are grateful to the experimental team (Wang, D.; Chen, K.; Xin, J.; Zhang, T.) for their efforts for the lake experiments.

**Conflicts of Interest:** The authors declare no conflict of interest.

## Appendix A

**Table A1.** Parameters in mathematical model.

| Parameter | Value | Unit | Description | Parameter | Value | Unit | Description |
|---|---|---|---|---|---|---|---|
| $m$ | 25 | kg | weight | $C_x(0)$ | −0.167 | — | longitudinal drag coefficient |
| $G$ | 245 | N | gravity | $C_Y^\alpha$ | 1.953 | — | vertical drag coefficient |
| $\Delta G$ | 0 | N | negative buoyancy | $C_Y^r$ | 0.226 | — | |
| $S$ | 0.042 | m² | projective area | $C_Z^\beta$ | −1.569 | — | lateral drag coefficient |
| $x_G$ | 0 | m | | $C_Z^p$ | 0 | — | |
| $y_G$ | −0.09 | m | center of gravity in body-fixed frame | $C_Z^q$ | −0.402 | — | |
| $z_G$ | 0 | m | | $C_R^\beta$ | 0 | — | roll moment coefficient |
| $L$ | 2 | m | length | $C_R^p$ | 0 | — | |
| $\rho$ | 1000 | kg/m³ | water density | $C_R^q$ | 0 | — | |
| $J_x$ | 0.065 | kg·m² | | $C_M^\beta$ | 0.748 | — | yaw moment coefficient |
| $J_y$ | 2.573 | kg·m² | moment of inertia | $C_M^p$ | 0.335 | — | |
| $J_z$ | 2.583 | kg·m² | | $C_M^q$ | −0.156 | — | |
| $\lambda_{11}$ | 2.561 | kg | | $C_N^\alpha$ | 0.692 | — | pitch moment coefficient |
| $\lambda_{22}$ | 39.219 | kg | additional mass | $C_N^r$ | −0.176 | — | |
| $\lambda_{33}$ | 33.408 | kg | | $K_{T1}$ | 0.193 | — | thrust coefficient of tail thruster |
| $\lambda_{26}$ | 1.792 | kg·m | additional static moment | $D_1$ | 0.071 | m | diameter of tail thruster |
| $\lambda_{35}$ | 1.792 | kg·m | | $K_{Q1}$ | −0.041 | — | torque coefficient of tail thruster |
| $\lambda_{44}$ | 0.824 | kg·m² | | $K_{T2}$ | 0.136 | — | thrust coefficient of tunnel thruster |
| $\lambda_{55}$ | 38.643 | kg·m² | additional moment of inertia | $D_2$ | 0.061 | m | diameter of tunnel thruster |
| $\lambda_{66}$ | 38.633 | kg·m² | | $K_{Q2}$ | −0.029 | — | torque coefficient of tunnel thruster |

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
