# Peer review of "Modified LOS Path Following Strategy of a Portable Modular AUV Based on Lateral Movement"

_jmse, doi:10.3390/jmse8090683_

Round 1
Reviewer 1 Report
The submission deals with a variant of LOS path following that is adapted to vehicles with lateral motion capabilities. I am not a deep expert in this specific area, but the strategy seems reasonable and the experimental results, including some trials in a lake, are good.
But regarding all statements with respect to the novelty of the ZFAUV, I would simply omit them. For example, right at the beginning of the abstract it is stated "The novel portable modular AUV (named ZFAUV) has the ability to move laterally". There are also many other modular, portable AUV that can move laterally, especially in research. Hence, the "novel" here seems somewhat misplaced. The same holds for statements like "Through the unique lateral movement of ZFAUV...".
For the different AUVs that are mentioned in the state of the art, it would be nice to have a table with a matrix of their actuation properties, i.e., number and type of thrusters and rudders, to have an easy overview.
When it is stated that "Naeem W. et al. reviewed some important guidance laws...", I suppose that this is meant to be a literature reference. This should be turned into a proper citation. Also, for some of the AUVs that are mentioned, the type and number of citations is not optimal - the authors should make sure that they use proper references.
The language is OK, i.e., everything is understandable, but it should be proof-read to check for bugs and to smooth the language. The math notations are weirdly formatted; there seems to be a strange shift up of all formulas, etc. Some abbreviations, especially LOS in the abstract, should be spelled out when they are introduced.
Reviewer 2 Report
REVIEW: Modified LOS path following strategy of a novel portable modular AUV based on lateral movement
Overall, the paper is well-structured and has potential for publication. However, some contents presented are not clear enough as below:
+ In the Introduction, the authors need to add some reference papers of kinds of control systems such as: X rudder, cross rudder, rudder behind propeller, rudder at front and so on. Also, many AUVs have over-actuated (the propellers of AUVs is over six propellers) were developed around the world. For the kinds of rudder behind propeller and AUV has 7 fixed propellers, the authors can refer as:
Vu, M.T.; Choi, H.S.; Nhat, T.Q.M.; Nguyen, N.D.; Lee, S.D.; Le, T.H.; Sur, J. Docking assessment algorithm for autonomous underwater vehicles. Appl. Ocean Res. 2020, 100, 102180.
Vu, M.T.; Choi, H.S.; Kang, J.I.; Ji, D.H.; Jeong, S.K. A study on hovering motion of the underwater vehicle with umbilical cable. Ocean Eng. 2017, 135, 137-157.
Vu, M.T., Van, M., Bui, D.H.P., Do, Q.T., Huynh, T.T., Lee, S.D., Choi, H.S., 2020. Study on Dynamic Behavior of Unmanned Surface Vehicle-Linked- Unmanned Underwater Vehicle System for Underwater Exploration. Sensors 20, 1329.
+ Almost equations and their notation of variables have to be arranged and revised again in order to meet the suitable format of the journal. The current forms of equations make the reader confused and hard to understand. Especially, Equation 1, 2 and all equations in section 4.3
+ The reference frames (earth-fixe and body-fixed frames) using in the paper were different from Fossen [7], thus it caused much confusion when reading paper. Normally, for underwater vehicle systems, z always represents the depth motion and has a downward direction (not y as shown in this paper). So, the reviewer recommends that the authors revise the reference frames and also some related equations.
+ In Section 2.2, the authors analyzed 7 motion cases of AUV, what happen if T3=T4<0 and T5=0, T3=T4<0 and T5 > or <5?
+ Equation 1 had 12 sub-equations. However, the model of AUV in horizontal plane includes 3 equations in kinematics and 3 equations in dynamic. The authors need to explain more about your equation. Also, in the last sub-equation of Eq. 1, T6 should be T5 ? Moreover, all the variables of equation 1 need to describe such as: V_T, beta, KT1, KQ1, D1 and so on.
+ If equation 2 expressed the model of AUV in horizontal plane, w should be v, and T5 should be T6 as shown in Figure 3. Also, what is l3?
+ The fuzzy PID controller needs to mention controller stability and fuzzy rule?
+ The variables in Figure 9 need to explain the meaning?
+ In the paragraph below Figure 10, “simulation experiments” word should be changed “simulation results” or “experimental results”?
+ Section 4.3 needs to arrange all equations.
+ The caption of Figure 5 was wrong. It was the same as Figure 23.
+ The reviewer recommends that the author need to prove the parameters of the controller (Ki, Kp, Kd) as well as the hydrodynamic coefficients of AUV? Besides, the figures of controller inputs such as the thruster forces or the force F in equation 8 have to include in the paper.
+ The contribution of conclusion is not mentioned clearly. The authors need to re-write the conclusion again to show the objective and contribution of this paper clearly. Also, future work should be added on Conclusion Section.
+The English writing of this paper should be thoroughly polished, especially some grammatical errors and formula format errors are required to be revised carefully.
I encourage authors to do these modifications.
Round 2
Reviewer 2 Report
The authors have solved previous comments well. It can be recommended for publication. However, the caption of Figure 25 and Figure 27 appeared twice. Please modify before publishing. In addition, the format of references needs to be unified with the aim to satisfy the requirement of the journal, and the doi number of some new references needs to be added in this paper. Please check carefully.